# Cognitive reserve is associated with education, social determinants, and cognitive outcomes among older American Indians in the Strong Heart Study
Astrid M. Suchy-Dicey [1] ✉, W. T. Longstreth Jr[2], Dedra S. Buchwald[2], Kristoffer Rhoads[2] & Thomas J. Grabowski[2]

Cognitive reserve, a component of resilience, may be conceptualized as the ability to overcome accumulating neuropathology and maintain healthy aging and function. However, research measuring and evaluating it in American Indians is needed. We recruited American Indians from 3 regional centers for longitudinal examinations (2010-13, n = 818; 2017-19, n = 403) including MRI, cognitive, clinical, and questionnaire data. We defined cognitive reserve by measuring the residual from individual regressions of cognitive tests over imaged brain volumes, adjusted for age and sex. Analyses examined three different metrics of cognitive reserve against sociodemographic, clinical, and longitudinal cognitive data in causal mediation models. Better cognitive reserve was significantly associated with more education, higher income, lower prevalence of depression, lower prevalence of diabetes, and lower prevalence of kidney disease, but we found no statistically significant evidence for an association with plasma biomarkers for Alzheimer's disease and related dementias, APOE e4 carrier status, alcohol use, body mass, or hypertension. Better cognitive reserve was associated with better cognitive function over mean 6.7 years follow-up (range 4-9 years); and the association for education with cognition over time was mediated in part (15-24%) by cognitive reserve. Cognitive reserve, although challenging to measure, appears important for understanding the range of cognitive aging in American Indians.

Cognitive resilience may be conceptualized using three different facets: as cognitive reserve, or resilience in cognition despite advancing pathological status; as brain reserve, or resilience in brain functionality despite advancing pathological status; or as brain maintenance, or resilience to change in neuropathologic status despite substantive risk factors such as advancing age[1]. Cognitive reserve is challenging to study—first because direct measures are not available, so operationalization using proxy and latent variables is still needed. Furthermore, in vivo measures of neuronal pathology are not widely available, further distancing available measures from reflecting true status. When neither the injury nor the mediating construct can be directly measured, conceptual frameworks can be difficult to construct and interpret.

Nevertheless, cognitive reserve, may be conceptualized as the ability to overcome the influence of accumulating brain pathology, in order to maintain cognition and functioning[2,3]. Thus, those with more cognitive reserve may be better equipped—perhaps via enhanced cortical thickness and/or connectivity—in compensating for accumulating disease burden, and thus preventing or delaying the emergence of clinical symptoms[4–7]. Such an abstract concept may be quantified in multiple ways: one option is to declare intact cognitive status, despite substantive accumulated pathology or advanced age. However, dichotomizing cognitive state as intact versus impaired may be overly reductionistic, arbitrary, and may introduce bias and measurement error in contexts where neuropsychological tests are not validated and normative data do not yet exist[8,9].

[1]Huntington Medical Research Institutes, Pasadena, CA, USA. [2]University of Washington, Seattle, WA, USA. ✉e-mail: astrid.suchy-dicey@hmri.org

A more ideal approach utilizes continuous estimation, thus also enabling adjustment for confounding by premorbid function[10]. In such an approach, cognitive reserve is represented as a continuous measure of association between selected measures of cognition and neuropathology, and could theoretically reflect those with abnormally positive aging, negative aging, or both[9]. Continuous estimation of cognitive reserve by regressing cognitive test scores against measures of imaged brain volumes, adjusting for confounders such as age and sex, and predicting the expected residual values from each regression provides an intraindividual metric of observed-over-expected performance[10]. As an estimate of continuous, comparative cognitive reserve, regressed residual metrics have unknown specificity to the chosen cognitive domain and pathologic marker[9].

The selection of specific measures to define the metric of cognitive reserve can influence the patterns of association for that metric. For example, a metric using a measure of processing speed over general brain volume might be demonstrative of global pathologies[1,10], whereas a metric of phonemic fluency over white matter hyperintensities (WMH) might be more sensitive to vascular pathologies[11], and a metric of memory over hippocampal loss might be more reflective of neurodegenerative pathologies[12].

Despite increasing interest in resilient aging and cognitive reserve, research is still needed to examine both risk and protective features in American Indians. Note that we use the term "American Indian" per specific direction from our partner Tribal communities and Community Advisory Board. American Indians convey many features associated with strength-based aging[13], including psychological resilience[14–16], community, and social support[17]. However, this population is also at particularly high risk for vascular disease, Alzheimer's disease (AD), and related dementias[18,19], in part due to the high burden of modifiable, clinical risk factors—although not all conventional AD risk factors appear generalizable in effect[20]. Thus, estimation of cognitive reserve in this unique population may help elucidate some of the observed, potentially modifiable effects from factors such as diabetes and depression[21–23], as well as some of the observed variance in mediating factors such as education. In previous research, we detected the strongest associations of brain, WMH, and hippocampal volumes with cognitive test performance[24], and thus structured candidate metrics using these imaging features, in combination with corresponding test scores for processing speed, fluency and executive function, and memory, respectively.

This study aimed to develop understanding of these metrics in this population facing unique challenges, with the ultimate goal to identify social determinants across the life-course that have the potential to promote healthier aging trajectories for American Indian older adults and their communities.

## Methods
### Setting
The Strong Heart Study (SHS) is a longitudinal study of American Indians from the Northern Plains, Southern Plains, and Southwest, recruited for multiple examinations between 1989–1999[25]. Using the SHS as a sampling frame, the ancillary Cerebrovascular Disease and its Consequences in American Indians study (CDCAI) conducted two neurological examinations in 818 consenting study survivors in 2010–2013 (Visit 1) and in 403 of the 818 in 2017–2019 (Visit 2), as previously reported[26].

### Ethics and inclusion
Institutional, Indian Health Service, and Tribal research review boards approved all procedures. All participants provided written, informed consent, and were compensated for their participation in this study. This study analysis was pre-proposed to the Strong Heart Study Publication & Presentations committee (Proposal 846, Approved 2024-06-13); a list of approved publications is available at https://strongheartstudy.org/Research/Papers-and-Abstracts/Approved-Papers-Dissertation. There was no additional preregistration for this study. All partnering Tribes and communities approved all written dissemination materials. Of note, the term "American Indian" is used herein, per specific direction from our study partner Tribal communities and Community Advisory Board, who have the final say in the representation of their identity. Terminology preferences vary substantively by Tribe, community, and individual, but this term is generally considered acceptable in the United States; is commonly the preferred term when individual Tribe or Tribal nations cannot be directly named; and is also the preferred term based on guidance from nationally-representative, self-governing, Indigenous-led organizations such as the National Congress of American Indians.

### Cognitive testing
The Weschler Adult Intelligence Scale 4th edition digit symbol coding subtest (WAIS-DSST, processing speed or visuomotor speed)[27] Controlled Oral Word Association FAS (COWA FAS, verbal fluency, executive function)[28], California Verbal Learning Test II Short Form (CVLT II-SF, verbal learning and memory)[29], were administered at both CDCAI Visit 1 and Visit 2. Modified Mini-Mental Status Examination (3MSE, general multidomain cognitive screening)[30], was also administered at both Visit 1 and Visit 2; other tests added at CDCAI Visit 2 included the National Alzheimer's Coordinating Center Uniform Data Set version 3.0 form C2 battery[31], with Montreal Cognitive Assessment (MoCA, multidomain general screening)[32], Number Span Test forward and backward (attention and working memory), Benson Complex Figure copy and recall (visuospatial skills and memory)[32,33], animal and vegetable naming tests (semantic fluency), Craft Story immediate and delayed recall (contextual verbal memory)[34], Trail Making Test A and B (processing speed, simple and divided attention)[35], and Multilingual Naming Test (MINT, semantic memory and naming)[36]. Of note, lower cognitive scores correspond with poorer cognition for all measures except the Trail Making timed tests wherein more time (seconds) needed to complete the activity corresponds to poorer performance. Internal consistency-reliability for cognitive tests in this population appear to follow a pattern of excellent fit and performance for overall or summary scores (omega coefficients >0.9), but poor metrics of fit and performance for individual domain, subcomponent, or tasks (omega <0.8;)[37–39]. Composite, multidomain cognitive test performance was also defined as the average of Z-scored cognitive tests.

### MRI
At both CDCAI Visit 1 and Visit 2, participants underwent 1.5T MRI scans with six sequences including sagittal T1-weighted localizer, co-registered 5 mm axial-T1, 5 mm axial-T2, 5 mm axial-T2* susceptibility-weighted images in the anterior commissure/posterior commissure plane, 3 mm axial fluid-attenuated inversion recovery (FLAIR) images, and 1.5 mm sagittal T1-weighted volumetric gradient echo. Additional details regarding repetition time, echo time, inversion time, slice thickness, slice spacing, acquisition matrix, number excitations, echo train, flip angle, and field of view have been provided in prior reports. MRI were processed for structural volumetrics, including total gray and white tissue for the whole brain, white matter hyperintensities (on FLAIR) or hypointensities (on T1), and hippocampus volume using version 5.3 of the FreeSurfer image analysis suite[23], with motion correction and visual checks at intermediate points in the process for gross misregistrations and other processing failures. Such problems were corrected manually. Only images that exceeded the field of view were censored, although these did not usually pass quality control. Skull stripping using cortical reconstruction by voxel-based parcellation of T1 images, skull stripping using cortical reconstruction, removal of non-brain tissue using a hybrid watershed-surface deformation procedure, automated Talairach transformation, segmentation of subcortical white matter and deep gray matter volumetric structures, intensity normalization, tessellation of the gray matter-white matter boundary, automated topology correction, and surface deformation following intensity gradients to place optimally the gray matter-white matter and gray matter-CSF borders at the location where the greatest shift in intensity defines the transition to the other tissue class. Overall, the results of these procedures include brain and cranial volumetric estimates that include cerebellum but not ventricles, CSF, and dura. All features were estimated in millimeters cubed, and then standardized to intracranial volume to account for interindividual variations in head size.

**Table 1 | Selected characteristics of American Indian participants aged 65–95 in the Strong Heart Study/ cerebrovascular disease and its consequences in American Indians (CDCAI) sub-cohort, at Visit 1 (2010–2013) and Visit 2 (2017–2019)**

| | Visit 1: 2010–13 N = 818 | Visit 2: 2017–19 N = 403 | P value |
|---|---|---|---|
| Age (years) | 73.0 (5.9) | 78.0 (4.7) | <0.001 |
| Female sex[c], % | 67.5% | 70.2% | 0.34 |
| Southwest | 12.0% | 12.8% | 0.80 |
| Northern Plains | 45.7% | 43.8% | |
| Southern Plains | 42.3% | 43.4% | |
| Education | | | 0.01 |
| Up to or some high school, % | 23.9% | 17.4% | |
| High school graduate or GED, % | 29.9% | 31.2% | |
| Some college, % | 30.4% | 32.9% | |
| Bachelor's degree or more, % | 15.6% | 17.2% | |
| Household annual income | | | 0.29 |
| <$10,000 per year | 30.5% | 24.8% | |
| $10–19,000 per year | 28.7% | 28.5% | |
| $20–34,000 per year | 21.4% | 22.1% | |
| $35,000+ per year | 19.3% | 21.6% | |
| Standard alcohol drinks per day[a] | 3.8 (7.1) | 2.0 (3.1) | 0.006 |
| Chronic kidney disease (CKD, eGFR <60 mL/min) | 26.9% | 51.0% | <0.001 |
| Overweight or Obese (BMI >25) | 84.7% | 81.4% | 0.15 |
| Hypertension | 80.8% | 81.8% | 0.68 |
| CESD score | 11.1 (8.7) | 12.7 (8.5) | 0.003 |
| Depressed (CESD score 16+), % | 24.3% | 30.7% | 0.02 |
| Fasting plasma glucose mg/dL | 123.8 (52.9) | 122.7 (54.9) | 0.74 |
| Diabetes mellitus, % | 49.5% | 47.9% | 0.61 |
| APOEe4 carrier | - | 20.9% | |
| pTau pg/mL | - | 6.3 (4.8) | |
| Ab42/40 ratio | - | 0.06 (0.01) | |
| GFAP pg/mL | - | 178.2 (96.0) | |
| NfL pg/mL | - | 41.0 (30.5) | |
| Weschler Adult Intelligence Scale 4th ed. digit symbol coding subtest (WAIS DSST) score | 44.8 (14.7) | 41.9 (14.1) | 0.02 |
| Brain volume, units: % intracranial vol | 67.2 (5.3) | 63.0 (6.1) | <0.001 |
| Residuals - WAIS DSST / brain volume(% IC), mean (range)[b] | 0 (-39.1, 42.6) | 0 (-37.6, 39.9) | >0.9 |
| Controlled Oral Word Association FAS (COWA) score | 24.4 (11.4) | 24.3 (11.1) | 0.92 |
| White matter hypointensities (WMH), units: % intracranial vol | 0.40 (0.36) | 0.26 (0.27) | <0.001 |
| Residuals - COWA / WMH volume (% IC), range[b] | 20.0 (-21.5, 64.9) | 15.5 (-20.0, 58.1) | <0.001 |
| California Verbal Learning Test II Short Form long delay free recall score (CVLT LF) | 5.5 (2.2) | 4.7 (2.4) | <0.001 |
| Hippocampus volume, units: % intracranial vol | 0.49 (0.05) | 0.48 (0.06) | <0.001 |
| Residuals – CVLT LF / Hipp. volume(% IC), range[b] | 0 (-6.6, 5.0) | 0 (-5.6, 5.4) | >0.9 |

**Table 1 (continued) | Selected characteristics of American Indian participants aged 65–95 in the Strong Heart Study/ cerebrovascular disease and its consequences in American Indians (CDCAI) sub-cohort, at Visit 1 (2010–2013) and Visit 2 (2017–2019)**

| | Visit 1: 2010–13 N = 818 | Visit 2: 2017–19 N = 403 | P value |
|---|---|---|---|
| Modified Mini-Mental Status Examination (3MSE) score | 88.5 (9.2) | 87.1 (9.5) | |
| Montreal Cognitive Assessment (MoCA) score | - | 18.9 (4.3) | |
| Animal naming test score | - | 13.8 (4.5) | |
| Vegetable naming test score | - | 9.6 (3.3) | |
| Craft Story immediate recall paraphrase score | - | 10.1 (4.3) | |
| Craft Story delayed recall paraphrase score | - | 8.8 (4.4) | |
| Benson Complex Figure copy score | - | 15.6 (1.7) | |
| Benson Complex Figure recall score | - | 8.8 (3.7) | |
| Number Span Test forward score | | 6.5 (2.3) | |
| Number Span Test backward score | | 4.5 (1.9) | |
| Trail Making Test A seconds | - | 65.4 (32.1) | |
| Trail Making Test B seconds | - | 169.4 (79.2) | |
| Multilingual Naming Test (MINT) score | - | 27.2 (3.2) | |
| Composite cognitive score (averaged Z-scores), range | −3.4, 1.7 | −1.8, 1.1 | |

[a]Alcohol standard drinks per day mean (SD) tabulated among users: Visit 1 n = 143 (17.5%), Visit 2 n = 66 (16.4%).
[b]Calculation of residual metrics using multiple variables with partial missingness resulted in slightly smaller numbers for these metrics of cognitive reserve: WAIS-Brain Visit 1 n = 765 and Visit 2 n = 298, COWA-WMH Visit 1 n = 765 and Visit 2 n = 298, CVLT-Hipp Visit 1 n = 744 and Visit 2 n = 289.
[c]Categorization of participants with self-identified male sex is mutually exclusive to those with female sex, per data collection methods and data completeness: % male at Visit 1 was 32.5% and at Visit 2 was 29.8%.
P value calculated based on ANOVA, Wilcoxon rank-sum, or T-test. Cognitive reserve operationalized as residuals of observed-over-expected performance in WAIS digit symbol test regressed over the degree of brain atrophy standardized to intracranial volume (adjusted for age and sex). Diabetes mellitus is categorized based on high fasting plasma glucose, prior diagnosis, or use of antihyperglycemics or insulin. Values are given as mean (SD) unless otherwise noted. Composite cognitive score is defined as the mean of Z-scores for all cognitive tests collected at each exam (Methods).
Terms defined: Weschler Adult Intelligence Scale Digit Symbol Coding Test (WAIS DSST) over brain volume as percent intracranial volume (WAIS-Brain), (middle) residuals of Controlled Oral Word Association f/a/s test (COWA) over white matter hypointensities volume (MRI T1 sequence) as percent intracranial volume (COWA-WMH), and (bottom) residuals of California Verbal Learning Test version II short form long delay free recall test (CVLT LF) over hippocampus volume as percent intracranial volume (CVLT-Hipp). Body mass index (BMI), Centers for Epidemiologic Studies Depression Scale (CESD), Apolipoprotein E (APOE), phosphorylated tau (pTau), amyloid beta 42/40 ratio (Ab42/40), glial fibrillary atrial protein (GFAP), neurofilament light chain (NfL).

## Other data

Participants self-reported age (years), sex (male; female), years of formal education, annual household income, and typical use of alcohol (drinks per day). Participants underwent clinical and anthropometric examinations, including basic blood and urine laboratory testing, medical history, and transcription of medications. Diabetes mellitus was defined as fasting blood glucose ≥126 mg/dL, prior physician diagnosis, or use of antihyperglycemic medications or insulin. Body mass index (BMI) was defined as weight in kilograms divided by height in meters squared, and categorized using standard criteria (normal BMI <25 kg/m²; overweight BMI 25 to 29.9; obese BMI 30+). Hypertension was defined as averaged, seated systolic blood

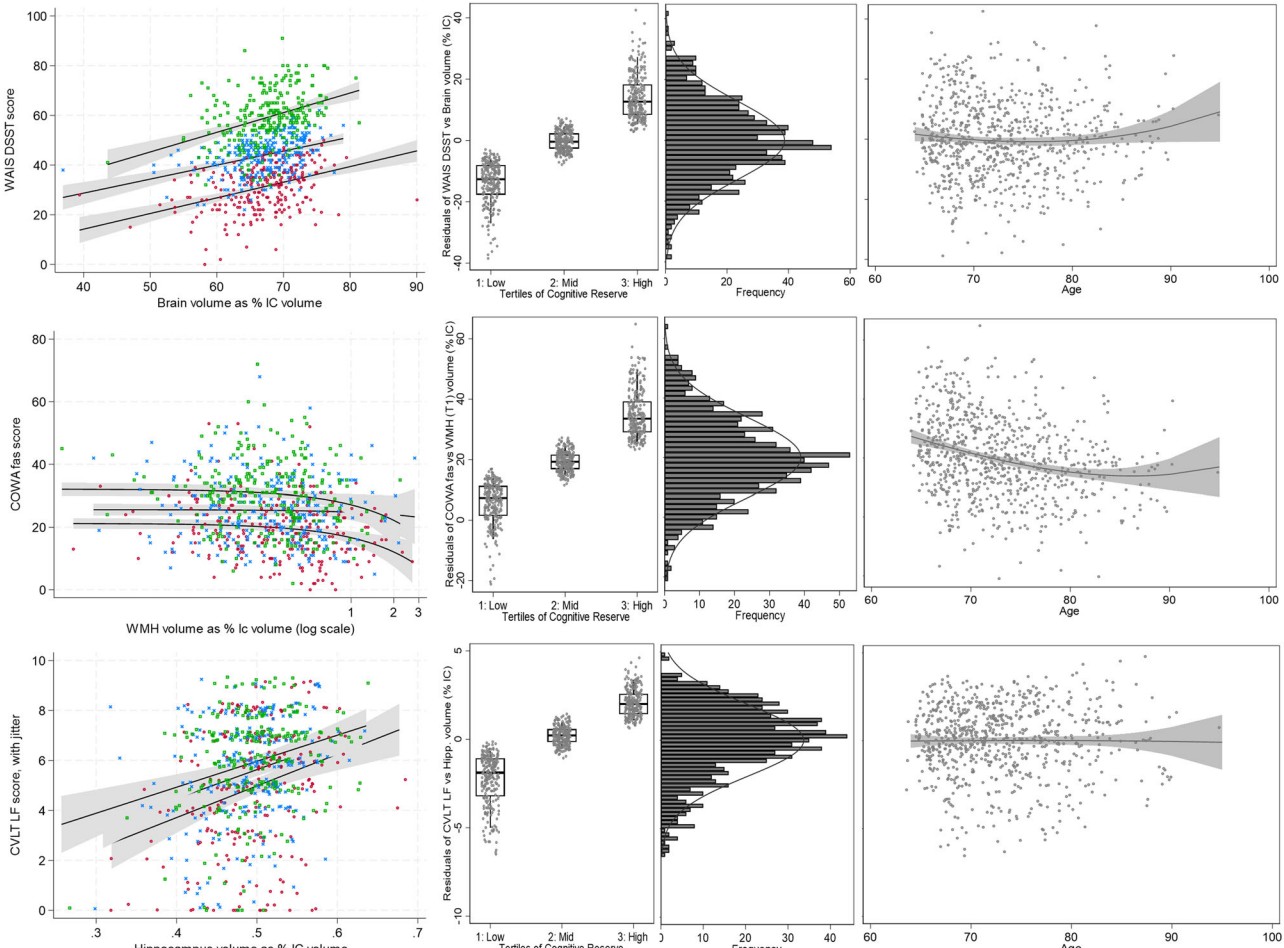

**Fig. 1 | Graphical representation of three metrics of cognitive reserve.** Graphical representation of three metrics of cognitive reserve, with (top) residuals of Weschler Adult Intelligence Scale Digit Symbol Coding Test (WAIS DSST) over brain volume as percent intracranial volume (WAIS-Brain), (middle) residuals of Controlled Oral Word Association f/a/s test (COWA) over white matter hypointensities volume (MRI T1 sequence) as percent intracranial volume (COWA-WMH), and (bottom) residuals of California Verbal Learning Test version II short form long delay free recall test (CVLT LF) over hippocampus volume as percent intracranial volume (CVLT-Hipp). Metrics of cognitive reserve defined by regression of cognitive test score over brain imaging feature, adjusted for age and sex. Left: scatterplots of cognitive test scores (Y-axis) by brain imaging features (X-axis). The coloration of scatter points with fit lines by tertile of metric for cognitive reserve, with the highest reserve in green, middle in blue, and lowest in red. Note: COWA-WMH scatterplot shown with X-axis on a log scale, and CVLT-Hipp plot with point-jitter for visibility purposes. Left-middle: Boxplots (mean, 25th, and 75th percentiles), with scatter overlay, and whisker bars (10th and 90th percentiles) for tertiles of (A, B, C) metrics of cognitive reserve. Right-middle: histograms for distribution of each metric with normal tracing. Right: scatterplots of metrics over age, with polynomial fit line and 95% confidence interval. Numbers included in the analysis: WAIS-Brain Visit 1 $n = 765$, COWA-WMH Visit 1 $n = 765$, CVLT-Hipp Visit 1 $n = 744$.

pressure ≥130 mmHg, diastolic ≥90, or the use of antihypertensive medications. Chronic kidney disease was defined as an estimated glomerular filtration rate <60 mL/min. Participants self-reported emotional and mental health using the 20-item Centers for Epidemiologic Studies Depression (CESD) symptoms scale, and depression was dichotomized using a cutoff ≥16. Plasma biomarkers related to Alzheimer's disease were measured (using samples from Visit 2) on the Quanterix platform including phosphorylated tau (pTau181), amyloid beta 42 and 40 (AB42/40 ratio), glial fibrillary acidic protein (GFAP), and neurofilament light chain (NfL). Apolipoprotein E epsilon 4 carrier status was assessed using standard genotyping.

## Cognitive reserve metrics
Cognitive reserve was operationalized as observed-over-expected cognitive performance, by regression of cognitive test score (WAIS-DSST score, COWA FAS overall score, or CVLT long delay free recall score) over a continuous estimate of brain pathology (total brain volume, WMH volume, or hippocampus volume, respectively; all standardized to IC volume); adjusted for age and sex, resulting in three metrics: WAIS-Brain, COWA-WMH, and CVLT-Hipp. Predicted values for each individual participant

according to the regression line were subtracted from their observed values, producing a "residual", or estimated degree of positive or negative cognition-beyond-pathology. In this paradigm, these residuals represent a continuous estimate for the latent variable of cognitive reserve, with positive values indicating higher reserve and negative values lower reserve. Residuals were then analyzed continuously, as well as in tertiles of low, middle, and high-reserve groups. This method was similar to those used by other groups, and the specific pairings (WAIS-Brain, COWA-WMH, CVLT-Hipp) were defined on the basis of theoretical associations of changes in cognition with associated affected region or pathology[10].

## Statistical analyses
Analyses were conducted as complete analysis, with exclusion only on the basis of missing data; for transparency, numbers included in each analysis figure and table were reported separately. Descriptive analyses estimated mean and standard deviation or count and percent, by examination visit, for participant characteristics, including the three metrics of cognitive reserve (WAIS-Brain, COWA-WMH, CVLT-Hipp). Graphical representations included histogram, scatterplot, linear fit, boxplot with whiskers, and

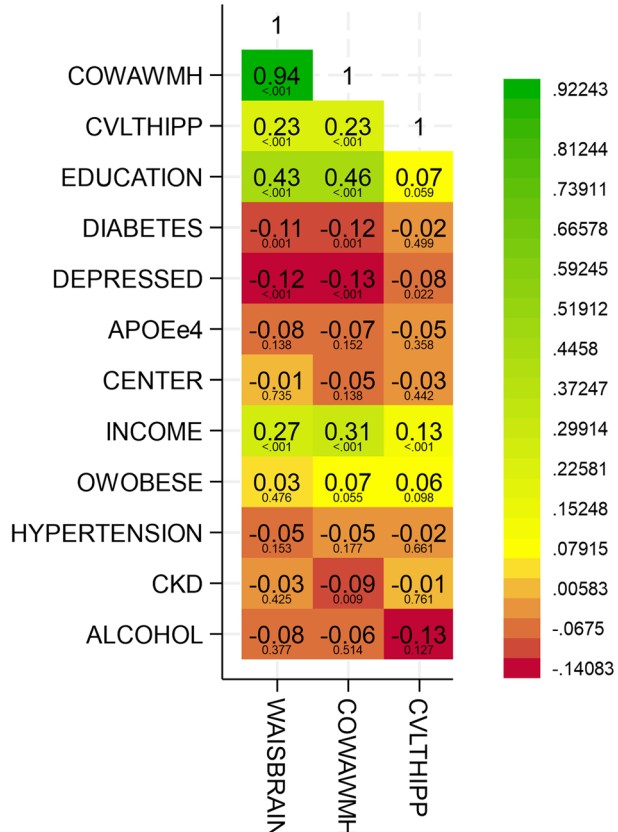

**Fig. 2 | Heatmap of associations between cognitive reserve metrics with sociodemographic and clinical characteristics.** Heatmap with coloration from red (strongest negative association) to green (strongest positive association), with yellow as intermediary values), and overlaid rho Pearson correlation coefficients and *P* values for associations of three metrics for cognitive reserve: residuals of Weschler Adult Intelligence Scale Digit Symbol Coding Test (WAIS DSST) over brain volume as percent intracranial volume (WAIS-Brain), residuals of Controlled Oral Word Association f/a/s test (COWA) over white matter hypointensities volume (MRI T1 sequence) as percent intracranial volume (COWA-WMH), and residuals of California Verbal Learning Test version II short form long delay free recall test (CVLT LF) over hippocampus volume as percent intracranial volume (CVLT-Hipp) with each other and with sociodemographic & clinical features. All metrics of cognitive reserve defined by regression of cognitive test score over brain imaging feature, adjusted for age and sex. Numbers included in the analysis: WAIS-Brain Visit 1 *n* = 765, COWA-WMH Visit 1 *n* = 765, CVLT-Hipp Visit 1 *n* = 744.

heatmap. Heatmap colorations were based on pairwise correlation coefficients (Pearson) and *P* values. Statistical tests for association included one-way ANOVA (F-test) for between-group differences, T-tests for equivalence in distributions, or regression. Causal mediation analysis estimated direct and indirect effects as well as the proportion of effect mediated for models with education, depression, or diabetes as exposure, composite cognitive test score as outcome, and metric of cognitive reserve as mediator. Evaluation of assumptions included assessment of normality in residuals, residual association with age and sex after adjustment, and heteroscedasticity. All analyses were done using Stata v17-18 (College Station, Texas) or R v4 (Vienna, Austria).

### Reporting summary

Further information on research design is available in the Nature Portfolio Reporting Summary linked to this article.

### Results

At Visit 1 (2010–13), this population-based cohort of American Indians had a mean age of 73, was majority female, and had about half with at least some

college (Table 1). Depression symptom (CESD) scores were high at both Visit 1 and Visit 2. Diabetes was detected in about half of the cohort at both visits. As noted in prior reports, income was relatively low, overweight and obesity, hypertension, and kidney disease were relatively common, and alcohol use was generally uncommon.

Metrics of cognitive reserve, defined as residuals of cognitive test scores over brain imaging features as % IC volume and adjusted for age and sex, included WAIS DSST over brain volume (WAIS-Brain), COWA fas over WMH volume (COWA-WMH), and CVLT LF over hippocampus volume (CVLT-Hipp). WAIS-Brain residuals were not significantly different comparing Visit 1 to Visit 2, over age, and were normally distributed (Fig. 1). COWA-WMH residuals were somewhat lower from Visit 1 to Visit 2 (mean residual difference 2.6 (95% CI: 1.5, 3.7; t-test P < 0.0001), significantly associated with age (beta -0.6; 95% CI: -0.5, -0.8; P < 0.0001), and were normally distributed. CVLT-Hipp residuals were not significantly different comparing Visit 1 and Visit 2, or by age, and were somewhat normally distributed but with a mild left skew.

Separating all three metrics of cognitive reserve into tertiles enabled comparison of low, middle, and high categories of cognitive reserve, independent of age and sex (Fig. 1). Graphically represented in boxplots by tertile or in scatterplots of each metric's cognitive test score (WAIS DSST, COWA fas, or CVLT LF, respectively) over brain imaged volumetric feature (overall volume, WMH volume, or hippocampus volume as % intracranial volumes, respectively), with low tertile colored red, middle in blue, and high in green, tertiles had clear separation and consistent linear associations for the WAIS-Brain metric, less clear separation but consistent linear associations for the CVLT-Hipp metric, and somewhat clear separation and inconsistent linear associations for the COWA-WMH metric.

Comparing in a heatmap these three metrics of cognitive reserve with sociodemographic, clinical, and psychological features (Fig. 2), better cognitive reserve (higher residual values) was significantly associated with more years of formal education (WAIS-Brain ρ = 0.45; 95% CI: 0.40, 0.51; *P* < 0.001 and COWA-WMH ρ = 0.48; 95% CI: 0.42, 0.53; *P* < 0.001), and not significant for CVLT-Hipp (ρ = 0.08; 95% CI: 0.01, 0.15; *P* = 0.06). Higher income was also significantly associated with the three metrics of cognitive reserve (WAIS-Brain ρ = 0.3, *P* < 0.001, COWA-WMH ρ = 0.3, *P* < 0.001, CVLT-Hipp ρ = 0.1, *P* < 0.001). Diabetes (WAIS-Brain ρ = −0.1 *P* = 0.001, COWA-WMH ρ = −0.1, *P* = 0.001) and depression (WAIS-Brain ρ = −0.1, *P* < 0.001, COWA-WMH ρ = −0.1, *P* < 0.001, CVLT-Hipp ρ = −0.1, *P* = 0.02) were significantly associated with lower cognitive reserve, although diabetes was not associated with CVLT-Hipp (ρ = 0.0, *P* = 0.5). APOEe4 allele status, field center, overweight/obesity, hypertension, chronic kidney disease (CKD), and alcohol use were not statistically associated with the three metrics of cognitive reserve. Boxplots with scatter overlay (Fig. 3) display the distribution of each of the three metrics of cognitive reserve, over four categories of education graphically suggested dose-dependent associations for WAIS-Brain and COWA-WMH metrics of cognitive reserve.

Using ANOVA to examine associations of tertiles of the three metrics of cognitive reserve—defined at Visit 1, with plasma biomarkers for Alzheimer's disease and related dementias—defined at Visit 2 (Table 2), only NfL had a significant association with COWA-WMH (tertile 1 to tertile 3 residual mean difference 10.2, *P* = 0.006). Of note, several of the comparisons showed significant departure from equal variance (i.e., significant Bartlett chi[2] test) across tertiles (categories) of cognitive reserve: WAIS-Brain metric with all four of the plasma biomarkers, COWA-WMH and CVLT-Hipp metrics with pTau and amyloid beta, and CVLT-Hipp with NfL.

Longitudinal regression analyses for tertiles of the 3 metrics of cognitive reserve—defined at Visit 1, with cognitive test scores—collected at Visit 2 (Table 3) detected generally consistent, significant associations of earlier cognitive reserve with later cognitive test performance. General cognition (3MSE; MoCA), processing speed (WAIS DSST) verbal fluency (COWA fas), semantic fluency (animal naming, vegetable naming, MINT), contextual memory (Craft story immediate and delayed recall), and executive function or attention (number span forward and backward, Trail Making

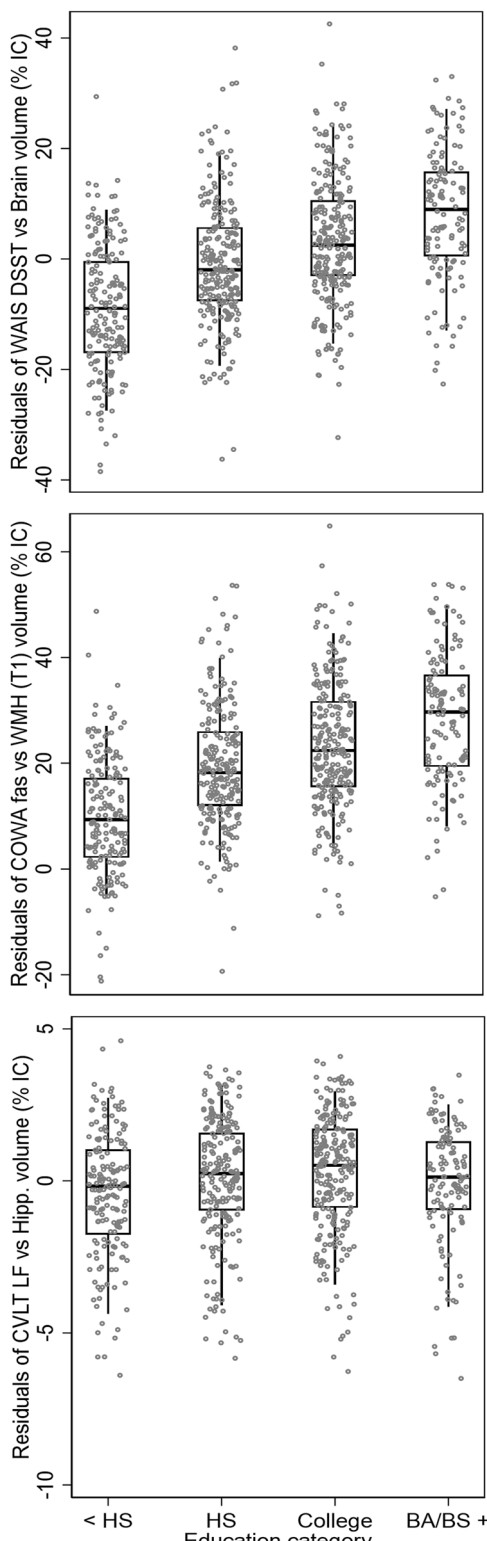

**Fig. 3 | Graphical figures comparing three metrics of cognitive reserve, over sociodemographic categories.** Boxplots (mean, 25th and 75th percentiles), with scatter overlay, and whisker bars (10th and 90th percentiles) for three metrics of cognitive reserve, over categories of education (less than high school, <HS; high school graduate, HS; any college; college graduate degree and beyond, BA/BS). Cognitive reserve metrics include (top) residuals of Weschler Adult Intelligence Scale Digit Symbol Coding Test (WAIS DSST) over brain volume as percent intracranial volume (WAIS-Brain), (middle) residuals of Controlled Oral Word Association f/a/s test (COWA) over white matter hypointensities volume (MRI T1 sequence) as percent intracranial volume (COWA-WMH), and (bottom) residuals of California Verbal Learning Test version II short form long delay free recall test (CVLT LF) over hippocampus volume as percent intracranial volume (CVLT-Hipp). All metrics of cognitive reserve defined by regression of cognitive test score over brain imaging feature, adjusted for age and sex. Numbers included in the analysis: WAIS-Brain Visit 1 $n = 765$, COWA-WMH Visit 1 $n = 765$, CVLT-Hipp Visit 1 $n = 744$.

mediation analysis examined the proportion of mediated effect for models with education, depression, or diabetes as exposure, metric of cognitive reserve—WAIS-Brain, COWA-WMH, or CVLT-Hipp—as mediator, and composite cognitive test score performance (Z-scale) as outcome, with adjustment for age and sex and robust standard errors estimation (Table 4). Of the three exposures considered, only education had a significant mediation (indirect) effect: accounting for an estimated 24% (95% CI: 5-44%, $P = 0.014$) of the WAIS-Brain metric association with later overall cognitive test performance, and accounting for estimated 15% (95% CI: 0–29%, $P = 0.045$) of the COWA-WMH metric association with later overall cognitive test performance.

## Discussion

Overall, these findings suggest that cognitive reserve—conceptualized as observed-over-expected cognitive performance and defined using three separate metrics—is positively associated with sociodemographic, clinical, psychological, and cognitive features of aging in American Indian older adults. This examination of cognitive reserve in American Indians compared cognitive reserve using 3 separate metrics: WAIS-Brain, a general metric reflecting processing speed and overall volumetric change; COWA-WMH, a vascular-sensitive metric reflecting changes in phonemic fluency with small vessel disease; and CVLT-Hipp, a memory-sensitive metric attuned to changes in a region targeted preferentially by Alzheimer's disease.

Income and depression were strongly and consistently associated with all three metrics of cognitive reserve (WAIS-Brain; COWA-WMH; CVLT-Hipp); education and diabetes are consistently associated with 2 of the metrics (WAIS-Brain, COWA-WMH). Kidney disease was associated only with COWA-WMH metric, as was the NfL plasma biomarker. In longitudinal analyses, all three metrics (WAIS-Brain; COWA-WMH; CVLT-Hipp) were also strongly related to cognitive test performance over time, in all cognitive domains except memory. Finally, causal analysis of education with cognitive test performance showed significant mediation (direct effect estimation) by WAIS-Brain (24%) and COWA-WMH (15%) metrics of cognitive reserve. Overall, these findings suggest that some metrics of cognitive reserve may be more informative than others, with patterns of association specific to their definition, but overall such metrics are generally consistent in associations with key social and clinical determinants, thus representing potentially modifiable protective processes in aging.

Our findings of strong and consistent associations for observed-over-expected performance with education, independent of age, are consistent with theoretical models that cognitive reserve is built over a lifetime, developed in early childhood through educational development in cognitive plasticity[40,41]. Such a model suggests that cognitive reserve, once built, may also be deconstructed through exposure to especial challenges such as socioeconomic deprivation[42] or chronic stress[43], also consistent with our findings of negative associations with depression. Aside from the processing speed-overall volume (WAIS-Brain) associations, the strongest associations were for executive function and phonemic fluency and small vessel injury

Test A and B) were all significantly associated with cognitive reserve measured a mean of 6.7 years earlier (range: 4–9 years). Analysis of visual processing and memory (Benson figure copy and recall) and verbal learning and memory (CVLT LF) scores were statistically associated with COWA-WMH and CVLT-Hipp metrics, but did not have statistical evidence of association with the WAIS-Brain metric.

Given positive associations of cognitive reserve with education, depression, and diabetes, as well as with cognitive test scores, causal

**Table 2 | Plasma biomarkers in American Indians age 70–95 in the Strong Heart Study/cerebrovascular disease and its consequences in American Indians (CDCAI) sub-cohort Visit 2 (2017-2019), over tertile- defined categories of high, middle, and low cognitive reserve at CDCAI Visit 1 (2010-2013)**

| Cognitive reserve metric | Characteristic value | Tertile1: Low resilience, range or mean (SD) | Tertile2: Mid resilience, range or mean (SD) | Tertile3: High resilience, range or mean (SD) | One-way ANOVA F-statistic (P value)[a] |
|---|---|---|---|---|---|
| WAIS DSST / Brain vol as % IC vol | Visit 1 cognitive residual | −39.1, −5.1 | −5.1, 4.9 | 4.9, 42.6 | |
| | Visit 2 pTau | 6.4 (4.4) | 9.0 (17.8) | 9.9 (37.9) | 0.54 (0.586)[b] |
| | Visit 2 AB4240 | 0.06 (0.02) | 0.06 (0.02) | 0.06 (0.01) | 1.00 (0.369)[b] |
| | Visit 2 GFAP | 174.5 (77.0) | 183.6 (107.0) | 172.0 (85.3) | 0.60 (0.547)[b] |
| | Visit 2 NfL | 42.8 (25.1) | 40.4 (24.3) | 39.7 (37.1) | 0.33 (0.716)[b] |
| COWA fas / WMH vol as % IC vol | Visit 1 cognitive residual, range | −21.5, 14.3 | 14.4, 24.8 | 24.8, 64.9 | |
| | Visit 2 pTau | 7.6 (13.3) | 7.0 (5.9) | 10.8 (39.1) | 0.86 (0.425)[b] |
| | Visit 2 AB4240 | 0.06 (0.01) | 0.06 (0.02) | 0.06 (0.01) | 0.41 (0.662)[b] |
| | Visit 2 GFAP | 175.6 (85.5) | 189.5 (98.1) | 165.6 (87.2) | 2.58 (0.077) |
| | Visit 2 NfL | 45.0 (26.3) | 44.8 (29.1) | 34.8 (32.0) | 5.22 (0.006) |
| CVLT LF / Hippocampus vol as % IC vol | Visit 1 cognitive residual, range | −6.6, −0.6 | −0.6, 1.0 | 1.0, 5.0 | |
| | Visit 2 pTau | 8.7 (16.9) | 7.2 (14.4) | 10.1 (38.7) | 0.38 (0.685)[b] |
| | Visit 2 AB4240 | 0.06 (0.01) | 0.06 (0.02) | 0.06 (0.01) | 0.55 (0.578)[b] |
| | Visit 2 GFAP | 168.2 (79.3) | 179.9 (91.1) | 175.4 (98.6) | 0.48 (0.617) |
| | Visit 2 NfL | 44.6 (38.2) | 40.3 (28.5) | 37.1 (21.2) | 1.96 (0.143)[b] |

[a]Numbers included in analysis: WAIS-Brain Visit 2 $n = 298$, COWA-WMH Visit 2 $n = 298$, CVLT-Hipp Visit 2 $n = 289$. Degrees of freedom between groups: 2, within groups: 378. $P$ value from one-way ANOVA of differences in means ($F$-test) in continuous values of plasma biomarkers, by tertile of cognitive reserve metric.
[b]Assumption not met of (null) Bartlett $P$ value for (chi2) test of equality of variances.
Cognitive reserve metrics defined as residuals of observed-over-expected performance in cognitive test score, regressed over brain image feature standardized to intracranial volume and adjusted for age and sex. Terms defined: Weschler Adult Intelligence Scale Digit Symbol Coding Test (WAIS DSST) over brain volume as percent intracranial volume (WAIS-Brain), (middle) residuals of Controlled Oral Word Association f/a/s test (COWA) over white matter hypointensities volume (MRI T1 sequence) as percent intracranial volume (COWA-WMH), and (bottom) residuals of California Verbal Learning Test version II short form long delay free recall test (CVLT LF) over hippocampus volume as percent intracranial volume (CVLT-Hipp). Analysis of Variance (ANOVA).

(COWA-WMH), especially with income and depression; associations for the metric related to memory (CVLT-Hipp) were not as strong or consistent as for the other two metrics.

APOEe4, ADRD plasma biomarkers, hypertension, body mass, field center (region), or alcohol use did not have detectable statistical evidence of association with metrics of cognitive reserve in these analyses. Other ways to conceptualize cognitive reserve may yield different results. Also, some of these results may reflect from lack of variance in the measures, such as for alcohol; with few users and low use among users, associations due to variability in alcohol use may be difficult to detect. However, education and income, depression, and diabetes—and to a lesser extent kidney disease—have consistently had strong associations with cognitive function, brain imaging and neuropathological findings, and other measures of ADRD in this population[20,23,26,44,45], which is consistent with these findings. It is possible that these are key variables to risk and resilience in this population. Future research should examine whether these association patterns are consistent for other Tribes and regions, other populations, and whether addressing the much higher prevalence of vascular disease and dementia may influence the balance of these associations for future generations[18,19].

Prior, cross-sectional research in non-Hispanic white Americans identified associations for a similarly-defined measure of cognitive reserve with cognitive test scores, with non-Alzheimer's disease pathology, and with APOEe4 genotype[10]. We found associations with education, depression, diabetes, and with other socioeconomic and clinical conditions. In longitudinal analyses, later cognition was associated with earlier cognitive reserve, independent of age, sex, baseline cognition, and time between examinations. Future research should examine associations with other neurological conditions.

## Limitations

Although causal sequence and temporality is not directly observable in cross-sectional associations, formative and environmental features such as education are likely to represent precursors to cognitive reserve, both

directly and indirectly. On the other hand, differential cognitive reserve may be either a contributor or an effect of disease comorbidities such as diabetes and depression. In our longitudinal analyses, cognitive reserve was a significant mediator of only the associations for education. However, two timepoints may not be adequate for answering causal longitudinal questions, and future research may benefit from evaluation of younger populations, more timepoints, and additional measures of cognitive reserve.

Our population included American Indian participants from multiple Tribes and communities in the United States Southwest, Southern Plains, and Northern Plains, but these findings may not generalize to other Tribal groups, regions, or generational cohorts. Cognitive tests may have differential validity across cultural groups, which can affect the generalizability of findings, although preliminary psychometrics in our study population (with more forthcoming) suggest similar test score interpretability as in other populations[37,38], with the exception that scores must be interpreted according to education and bilingual experience. Discriminant properties of cognitive test score cutoffs, in particular, require adjustment and validation for appropriate diagnostics—both for impaired, as well as for evaluation of "super-agers", or those who maintain healthy cognition despite advanced age and irrespective of pathology status. Also, due to the selection structure of this longitudinal cohort, our findings may be subject to an increased likelihood of Type II error if the outcome (e.g., cognitive resilience) is unduly influenced by the likelihood of survival to participation over Visits, although the metrics evaluated appear similar between the two visits. Finally, other types of resilience, such as psychological or community resilience have yet to be explored in this context. Ongoing research in these communities aims to provide insights into this area in the future.

## Conclusion

In summary, this description of cognitive reserve in American Indians suggests that cognitive reserve is strongly associated with education, other socioeconomic factors, depression, diabetes and other clinical factors, and

**Table 3 | Cognitive test scores in American Indians age 70–95 in the Strong Heart Study/cerebrovascular disease and its consequences in American Indians (CDCAI) sub-cohort Visit 2 (2017–2019), over tertile-defined categories of high, middle, and low cognitive reserve at CDCAI Visit 1 (2010–2013)**

| | WAIS-Brain | | COWA-WMH | | CVLT-Hipp | |
|---|---|---|---|---|---|---|
| | One-way ANOVA *F*-statistic[a] | *P* value | One-way ANOVA *F*-statistic[a] | *P* value | Oneway ANOVA *F*-statistic[a] | *P* value |
| 3MSE (Visit 2) | 17.06 | <0.0001[b] | 36.06 | <0.0001[b] | 21.05 | <0.0001[b] |
| WAIS (Visit 2) | 90.13 | - | 139.65 | <0.0001 [b] | 7.60 | 0.0006 |
| COWA (Visit 2) | 13.45 | <0.0001[b] | 27.80 | - | 4.86 | 0.0082 |
| CVLT long free (Visit 2) | 1.71 | 0.1831 | 4.01 | 0.0189 | 34.68 | - |
| MoCA (Visit 2) | 18.05 | <0.0001[b] | 33.96 | <0.0001[b] | 19.07 | <0.0001 |
| Animal naming (Visit 2) | 7.07 | 0.0010 | 12.42 | <0.0001 | 3.56 | 0.0295 |
| Vegetable naming (Visit 2) | 8.11 | 0.0004 | 12.10 | <0.0001 | 14.98 | <0.0001 |
| Craft immediate recall (Visit 2) | 11.70 | <0.0001 | 14.01 | <0.0001 | 3.73 | 0.0250 |
| Craft delayed recall (Visit 2) | 9.13 | <0.0001 | 10.29 | <0.0001 | 5.89 | 0.0030 |
| Benson figure copy (Visit 2) | 2.72 | 0.0670[b] | 5.26 | 0.0056 | 3.46 | 0.0326[b] |
| Benson figure recall (Visit 2) | 1.08 | 0.3414 | 3.26 | 0.0394 | 15.75 | <0.0001 |
| Number span forward (Visit 2) | 5.51 | 0.0044 | 12.05 | <0.0001 | 0.05 | 0.9518 |
| Number span backward (Visit 2) | 15.17 | <0.0001 | 26.19 | <0.0001[b] | 3.18 | 0.0428 |
| Trails A seconds (Visit 2) | 17.21 | <0.0001[b] | 28.85 | <0.0001[b] | 3.30 | 0.0382[b] |
| Trails B seconds (Visit 2) | 4.75 | 0.0092 | 5.17 | 0.0061[b] | 0.04 | 0.9602[b] |
| MINT (Visit 2) | 21.42 | <0.0001[b] | 23.55 | <0.0001[b] | 5.79 | 0.0034[b] |

[a]Numbers included in the analysis: WAIS-Brain both Visit 1 and Visit 2 $n$ = 298, COWA-WMH both Visit 1 and Visit 2 $n$ = 298. Degrees of freedom between groups: 2, within groups: 378. CVLT-Hipp both Visit 1 and Visit 2 $n$ = 289. $P$ value from one-way ANOVA of differences in means ($F$-test) in continuous values of plasma biomarkers, by tertile of cognitive reserve metric.
[b]Assumption not met of (null) Bartlett $P$ value for (chi2) test of equality of variances.
Cognitive reserve metrics defined as residuals of observed-over-expected performance in cognitive test score, regressed over brain image feature standardized to intracranial volume and adjusted for age and sex. Assumption not met of null Bartlett $P$ value for (chi2) test of equality of variances. Additional adjustments for follow-up time or with standardization of test scores over time elapsed provided similar results. Composite cognitive test performance, defined as an average of Z-scored tests by visit, also provided similar patterns of association. Terms defined: Weschler Adult Intelligence Scale Digit Symbol Coding Test (WAIS DSST) over brain volume as percent intracranial volume (WAIS-Brain), (middle) residuals of Controlled Oral Word Association f/a/s test (COWA) over white matter hypointensities volume (MRI T1 sequence) as percent intracranial volume (COWA-WMH), and (bottom) residuals of California Verbal Learning Test version II short form long delay free recall test (CVLT LF) over hippocampus volume as percent intracranial volume (CVLT-Hipp). Modified Mini-Mental State Examination (3MSE), Montreal Cognitive Assessment (MoCA), Multilingual Naming Test (MINT).

**Table 4 | Causal mediation analysis of sociodemographic or clinical characteristic exposures and cognitive reserve metric mediators, in the Strong Heart Study/cerebrovascular disease and its consequences in American Indians (CDCAI) sub-cohort at Visit 1 (2010–2013), on composite cognitive test scores (Z-scale), defined at CDCAI Visit 2 (2017–2019)**

| Mediator | Exposure | Total effect beta (95% CI) | Direct effect beta (95% CI) | Indirect effect beta (95% CI) | Proportion mediated (95% CI) | *P* value |
|---|---|---|---|---|---|---|
| WAIS-Brain | Education | 0.39 (0.26, 0.51) | 0.29 (0.15, 0.43) | 0.09 (0.03, 0.16) | 0.24 (0.05, 0.44) | 0.014 |
| | Depression | | 0.38 (0.25, 0.51) | 0.01 (−0.01, 0.04) | 0.04 (−0.03, 0.10) | 0.260 |
| | Diabetes | | 0.38 (0.26, 0.51) | 0.0 (−0.01, 0.01) | 0.0 (−0.01, 0.01) | 0.984 |
| COWA-WMH | Education | 0.43 (0.29, 0.56) | 0.37 (0.22, 0.51) | 0.06 (0.00, 0.12) | 0.15 (0.00, 0.29) | 0.045 |
| | Depression | | 0.41 (0.28, 0.55) | 0.01 (0.00, 0.03) | 0.02 (−0.02, 0.06) | 0.374 |
| | Diabetes | | 0.42 (0.29, 0.56) | 0.00 (−0.01, 0.02) | 0.01 (−0.02, 0.04) | 0.552 |
| CVLT-Hipp | Education | 0.40 (0.27, 0.52) | 0.39 (0.28, 0.51) | 0.01 (−0.03, 0.04) | 0.02 (−0.07, 0.10) | 0.679 |
| | Depression | | 0.40 (0.27, 0.53) | −0.01 (−0.03, 0.01) | [a] | [a] |
| | Diabetes | | 0.40 (0.27, 0.52) | 0.0 (−0.01, 0.01) | 0.0 (−0.02, 0.03) | 0.676 |

[a]When direct and indirect coefficient signs are opposite (one positive, one negative), the interpretation of proportion mediated is unclear.
Numbers included in analysis: WAIS-Brain both Visit 1 and Visit 2 $n$ = 298, COWA-WMH both Visit 1 and Visit 2 $n$ = 298. Model-directed acyclic flows include exposure (e.g., diabetes at Visit 1), mediator (metric of cognitive reserve, e.g., WAIS-Brain volume, at Visit 1), and outcome (e.g., composite cognitive score at Visit 2). Total effect is an association of exposure to outcome (direct + indirect); direct = excljuding effect of mediator, indirect = effect via mediator. Assumes causality. Direct effect is the association of exposure (diabetes or depression) on outcome (composite cognitive test score) without the mediator (cognitive reserve); indirect effect is the effect of exposure on outcome estimated to function through the mediator.
Terms defined: Weschler Adult Intelligence Scale Digit Symbol Coding Test (WAIS DSST) over brain volume as percent intracranial volume (WAIS-Brain), (middle) residuals of Controlled Oral Word Association f/a/s test (COWA) over white matter hypointensities volume (MRI T1 sequence) as percent intracranial volume (COWA-WMH), and (bottom) residuals of California Verbal Learning Test version II short form long delay free recall test (CVLT LF) over hippocampus volume as percent intracranial volume (CVLT-Hipp).

later cognitive changes. This work also suggests that some metrics of cognitive reserve (e.g., residuals of processing speed over total brain volume or residuals of verbal fluency over white matter lesion burden) may be better markers or determinants of overall cognitive reserve than others (e.g., residuals of memory over hippocampal volume), although each metric has unique and specific association patterns. This work has the potential to inform understanding of risk and prevention for Alzheimer's disease and related dementias.

## Data availability

Data from the Strong Heart Study and its ancillary studies cannot be made publicly available, but can be accessed—per study and Tribal policies—as described on the Strong Heart Study website: https://strongheartstudy.org.

## Code availability

Statistical code for these analyses is available at https://github.com/astridsd/CommPsychol2025.

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

## Acknowledgements

The authors wish to thank all Strong Heart Study staff, participants, and communities. This study has been funded in whole or in part with federal funds from the National Institutes of Health, including K01AG057821, P50AG005136, R01HL093086, and R01AG070822. The Strong Heart Study has been funded in whole or in part with federal funds from the National Heart, Lung, and Blood Institute under contract numbers 75N92019D00027, 75N92019D00028, 75N92019D00029, and 75N92019D00030. The study was previously supported by research grants: R01HL109315, R01HL109301, R01HL109284, R01HL109282, and R01HL109319 and by cooperative agreements: U01HL41642, U01HL41652, U01HL41654, U01HL65520, and U01HL65521. The funders had no role in study design, data collection and analysis, the decision to publish, or the preparation of the manuscript. The opinions expressed in this paper are solely the responsibility of the author(s) and do not necessarily reflect the official views of the Indian Health Service or the National Institutes of Health.

## Author contributions

All authors (A.M.S.-D., T.J.G., W.T.L., K.R., D.S.B.) contributed to data colleciton; A.M.S.-D. and T.J.G. conceptualized hypotheses; A.M.S.-D., W.T.L., and K.R. finalized analytic models; A.M.S.-D. conducted analyses; A.M.S.-D. wrote the manuscript; all authors (A.M.S.-D., T.J.G., W.T.L., K.R., and D.S.B.) contributed to manuscript editing and results interpretation.

## Competing interests

The authors (A.M.S.-D., T.J.G., W.T.L., K.R., and D.S.B.) declare no competing interests.
