## [Transparent Peer Review file · Communications Psychology]

Cognitive reserve is associated with education, social determinants, and longitudinal cognitive change but not with conventional Alzheimer's disease biomarkers and risk factors among older American Indians: data from the Strong Heart Study

Corresponding Author: Dr Astrid Suchy-Dicey

Version 0:

Decision Letter:

Dear Dr Suchy-Dicey,

Thank you for your patience during the peer-review process. Your manuscript titled "Cognitive reserve, comorbidities, and cognitive change among older American Indians: The Strong Heart Study" has now been seen by 2 reviewers, and I include their comments at the end of this message. They find your work of interest but raised some important points. We are interested in the possibility of publishing your study in *Communications Psychology*, but would like to consider your responses to these concerns and assess a revised manuscript before we make a final decision on publication.

We therefore invite you to revise and resubmit your manuscript, along with a point-by-point response to the reviewers. Please highlight all changes in the manuscript text file.

Editorially, we consider it important that the revised manuscript addresses the request from both reviewers for evidence validating the approach to operationalizing resilience. Please run additional regression models as suggested by Reviewer 1 as well as provide additional supporting evidence from the existing literature (if available) as suggested by Reviewer 2.

Please confirm and state in the manuscript that your work was approved by SHS along the lines of "review and approval was received from the Strong Heart Steering Committee and Strong Heart Study participating nations/tribes". Please add an Ethics and Inclusion Statement (<https://www.nature.com/commpsychol/editorial-policies/authorship#authorship-inclusion-and-ethics-in-global-research>) to your Methods section.

Please report the language/terminology used in data collection (i.e., "Native American" or "American Indian") in the Methods. As long as the terminology that was used in collecting the data is reported, you are welcome to change the terminology used in the manuscript in line with reviewer feedback and following APA guidelines, which also recommends using the name that the people use to refer to themselves. Regardless of whether you undertake a revision in terminology, we ask that you continue to provide a rationale for your choice of terminology in the text so that our readers appreciate that this was a thoughtful decision.

Please also report the peoples or nations that the participants belonged to in the Methods.

Please ensure you follow our statistical guidelines when reporting statistics (<https://www.nature.com/commpsychol/submit/submission-guidelines#statistical-guidelines>). Please note in particular our requirements for the reporting and interpretation of null-results. Non-significant findings derived from null-hypotheses significance tests should be reported in full, but may not be interpreted. Where you interpret null results, this interpretation must be based on Bayes Factors or equivalence tests.

I am attaching an Editorial Requests Table that details critical reporting requirements for the revised manuscript. Please attend to each item and ensure your manuscript is fully compliant. We are requesting that your manuscript aligns with these requirements as this facilitates the evaluation of your manuscript, reducing delays in re-review and potential future acceptance. If your revised manuscript is not aligned with these requests on major issues, such as those concerning statistics, it may be returned to you for further revisions without re-review. Additional information can be found in our style and formatting guide <https://www.nature.com/documents/commspsychol-style-formatting-guide-accept.pdf>>Communications Psychology formatting guide.

Please use the following link to submit your

- revised manuscript,
- point-by-point response to the referees' comments,
- cover letter (as a separate document),
- the Editorial Policy Checklist (see below),
- the Reporting Summary (see below), and
- the completed Editorial Request Table (attached):

Link Redacted

Best regards,

Jennifer Bellingtier

on behalf of
Claudia von Bastian, PhD
Editorial Board Member
Communications Psychology
orcid.org/0000-0002-0667-2460

REVIEWER EXPERTISE:

Reviewer #1 cohort/longitudinal studies, cognitive aging & dementia
Reviewer #2 cognitive reserve, cognitive aging & dementia

REVIEWER REPORTS:

Reviewer #1 (Remarks to the Author):

Overall

The manuscript presents original research on a novel topic that merits publication and is of interest to the scientific community. A particular strength of the study was that cognitive reserve was investigated in native Americans who represent an under-researched population. However, the paper shows clear room for improvement with respect to applying rigorous methodology, following reporting standards, and explicitly stating limitations to generalizability. I therefore recommend the following changes:

Major

The authors mention in the introduction that their estimate of resilience was "specific to the chosen cognitive domain and pathologic marker". However, the issue of validating the residuals from the regression model is not addressed in the paper. Cross-validating the residual approach towards operationalizing resilience would very much strengthen the manuscript and substantiate the findings. I would therefore suggest to derive several residual metrics indicative of cognitive reserve using other cognitive domains and pathologic markers, and then replicate the analyses presented in the paper. For example, years of formal education or executive function from the trail-making test can be used as another indicator of cognitive performance. In the MRI data, hippocampal atrophy, white matter lesions on the structural MRI data, and ventricular volumes

can be used as alternative pathologic markers. Previous research demonstrated that these MRI markers showed stronger associations with cognitive decline, progression towards dementia, and dementia risk. They are therefore better outcome variables for the regression model as the MRI total brain volume, and may be more sensitive a neuronal correlate of cognitive decline. Adding these additional regression models would help to cross-validate the cognitive resilience construct and strengthen the findings of the present study.

Minor

The term "American Indians" should be avoided to follow APA 7 guidelines on the usage of bias-free language in research. Bias-free alternative ethnic descriptors are "Native Americans", "Native North Americans" or "indigenous people of (North) America". The authors justify the use of "the term "American Indian" [...] in accordance with contemporary US legal structures". However, this justification is insufficient for a medical science publication because US legal terms are not reflective of bias-free language use and the current knowledge about ethnicity in a research context. For example, intersectionality is an important concept in research on ethnic groups, but it does not exist in the US legal system as seen in the DeGraffenreid v. General Motors case. The language used by contemporary US legal structures is not suitable for research publications. More information on bias-free language can be found here: [https://libguides.massgeneral.org/APA7/biasfree-language#:~:text=In general%2C refer to an,avoid the term "Indian"](https://libguides.massgeneral.org/APA7/biasfree-language#:~:text=In general%2C refer to an,avoid the term)

The Wechsler Adult Intelligence Scale is not a culture-free intelligence test and may be biased when carrying out research in Native Americans. Since all participants were Native Americans, this bias is unlikely to affect the findings, because the bias mainly becomes apparent when comparing Native Americans with European Americans. Nevertheless, the culture bias of the test should be mentioned in the limitations section along with the need for further research using culture-free tests.

Estimates of reliability (internal consistency): A report of the internal consistency of the WAIS scale and the other psychometric data used is missing. At least, Cronbach's alpha should be reported at the beginning of the results section. It is important to know the reliability estimates from the sample because the tests were used in the Native American population but constructed and validated in the general population. Similarly, an internal consistency metric for the CESD should be reported due to differences in depression across various ethnic groups.

BMI: The BMI was categorized into normal, overweight, and obese. It would be interesting to know if any participants were underweight with a BMI less than 18.5. Previous research showed that being underweight was associated with cognitive impairment in late life.

Alcohol: The plot (Supplementary Figure 4) shows alcohol has a skewed distribution with only a few participants consuming more than 7 drinks per day. This limits the generalizability of the non-significant effect of alcohol consumption on cognitive reserve and should be mentioned as a limitation.

"Possibly other measurements of cognitive reserve, either using different cognitive and pathologic measures or different mathematic constructs, may provide a better map for this esoteric construct." "Esoteric construct" is not a good word choice here, I would replace this with "operationalizing this unobserved, latent construct."

"On the other hand, better cognitive reserve may be either a contributor or an effect of disease comorbidities such as diabetes and depression." Should this sentence read: "lower cognitive reserve."

Table 4: The individual component paths of the mediation model should be reported as well: The path from the exposure to the mediator (Diabetes V1 à Cognitive V1) and mediator to the outcome (Cognitive reserve V1 -> Composite cognitive score V2). These paths can explain why the mediation was not significant.

Reviewer #2 (Remarks to the Author):

ABSTRACT

1. In lines 3-4, the authors write, "However cognitive resilience has not been measured or evaluated in American Indian elders." Please clearly state the purpose of the investigation, e.g., "The purpose of this investigation is to evaluate factors associated with cognitive reserve or cognitive resilience in American Indian elders."
2. Cognitive reserve and cognitive resilience are conceptually different but the authors seem to use them interchangeably. Please use one term to avoid confusing readers.

INTRODUCTION/BACKGROUND

1. The state of the field is clearly expressed in this section, which justifies the importance of the research question.
2. The concepts of cognitive reserve and resilience overlap, so this section warrants more explanation to avoid confusing readers.
3. Please consider clearly defining cognitive reserve and cognitive resilience per the Collaboratory on Research Definitions for Reserve and Resilience on Aging (PMID: 36653245) earlier in the manuscript. It is referenced in the discussion. An upfront and early delineation of the concepts can improve the reader's comprehension of the paper.
4. The introduction mentions that American Indian persons are at increased AD risk, but the authors do not quantify that risk with a rate.
5. If persons aged 80 and older are in the sample, it would be interesting to use cognitive testing to determine what proportion meets criteria consistent with super-agers.

METHODS

1. The selection of study participants and study variables utilized for the investigation are clear.
2. Is using regression of WAIS-DCS scores over standardized total brain volume estimates to operationalize cognitive reserve a proven approach? The paper cited used and validated the method in the context of cognitive resilience. Again, be cautious about how the terms “reserve” and “resilience” are used. Please make this point clear so as not to confuse readers. If the approach represents a novel use for cognitive reserve, is data supporting its validity in that context?

RESULTS

1. The data are presented appropriately, but it needs to be clarified which results are statistically significant in Table 1.
2. Define what “more formal education” means (line 121). Are you referring to years of education?
3. Please use spelling and grammar-checking software to detect misspelled words, “Both diabetes and depression had significant direct effect ($P < 0.0001$)...” (line 145).

DISCUSSION

1. Be clear that cognitive reserve and cognitive resilience are conceptually different. In lines 158 to 160, the following sentence suggests they are interchangeable: “Cognitive reserve, or cognitive resilience to biological or pathological changes, is challenging to study—first because as a theoretical, and direct measures are not available, so operationalization of proxy and latent variables is necessary.”
2. If the sample reduction between Visit 1 and Visit 2 is unrelated to selective survival, please comment on other possibilities.

OVERALL

1. The authors address an important topic in cognitive, i.e., factors associated with resilience in an underrepresented research population.
2. Exploring cognitive risk and protective factors in diverse samples can yield critical insights into disease manifestation, treatment, and prevention. Please consider citing the Alzheimer’s Association Facts and Figures reports that discuss AD/DR research samples that do not match the country’s general population to highlight your work’s importance further.
3. The longitudinal design is a strength of this work.
4. The authors alternate between adjectival and noun forms, i.e., American Indians vs. American Indian elders. Use the same form per the journal’s guidance for consistency.
5. Overall, the paper is important and timely to the field. There are a few grammatical errors. Spelling and grammar-checking software is a consideration for catching errors and improving overall readability. The word, cognitive, is misspelled in the abstract: “Analyses examined residuals of cognitive reserve against sociodemographic, clinical, and longitudinal cognitive data in causal mediation models.”

EDITORIAL POLICIES

We ask that you ensure your manuscript complies with our editorial policies and reporting requirements.

To that end, we require revised manuscripts to be accompanied by two completed items: a reporting summary that collects information on study design and procedure, and an editorial policy checklist that verifies compliance with all required editorial policies.

- <https://www.nature.com/documents/nr-reporting-summary.zip>>Nature Research Reporting Summary
- <https://www.nature.com/documents/nr-editorial-policy-checklist.pdf>>Editorial Policy Checklist

All points on the policy checklist must be addressed. Your revised manuscript can only be sent back to the referees if these checklists are completed and uploaded with the revision.

Notes: If you have submitted a Stage 1 Registered Report, Review, Primer, Comment, or Perspective you do not need to submit these forms. If you have already submitted these forms, you may disregard this request.

** Visit Nature Research's author and referees' website at <http://www.nature.com/authors>>www.nature.com/authors for information about policies, services and author benefits**

Version 1:

Decision Letter:

Dear Dr Suchy-Dicey,

Your manuscript titled "Cognitive reserve, comorbidities, and cognitive change among older American Indians: The Strong Heart Study" has now been re-reviewed. Because the original reviewers were unavailable, we invited an expert with the same expertise to evaluate your work and the revisions you undertook in response to the previous reports. In light of their advice I am delighted to say that we are happy, in principle, to publish a suitably revised version in Communications Psychology.

We therefore invite you to revise your paper one last time to address the remaining concerns of our reviewer and a list of editorial requests. At the same time we ask that you edit your manuscript to comply with our format requirements and to maximise the accessibility and therefore the impact of your work.

EDITORIAL REQUESTS:

SUBMISSION INFORMATION:

OPEN ACCESS:

*** DATA AVAILABILITY:**

Link Redacted

Best regards,

Jennifer Bellingtier

Jennifer Bellingtier, PhD
Senior Editor
Communications Psychology

REVIEWERS' EXPERTISE:

Reviewer #3 cognitive aging & dementia, longitudinal studies

REVIEWERS' COMMENTS:

Reviewer #3 (Remarks to the Author):

Disclaimer: I was contacted to provide comments on authors' response to original reviewers (I was not one of them). I read the manuscript as well as the rebuttal letter.

In my assessment, authors provided a satisfactory response to original reviewers' comments. Indeed, the comments were very much on point, and tackled important issues that warranted attention.

Particularly important is the fact that authors now consider alternative specifications of reserve deriving from more explicit sources of cerebral pathology (vascular damage, atrophy), as opposed to in relation to total brain volume, which is a rather crude measure, conflating non-pathological differences in brain structure and pathological insults.

In light of these revisions, I think the paper is now much improved and can be acceptable for publication.

Reviewer #1 (Remarks to the Author):

Overall

The manuscript presents original research on a novel topic that merits publication and is of interest to the scientific community. A particular strength of the study was that cognitive reserve was investigated in native Americans who represent an under-researched population. However, the paper shows clear room for improvement with respect to applying rigorous methodology, following reporting standards, and explicitly stating limitations to generalizability. I therefore recommend the following changes:

Major

The authors mention in the introduction that their estimate of resilience was “specific to the chosen cognitive domain and pathologic marker”. However, the issue of validating the residuals from the regression model is not addressed in the paper. Cross-validating the residual approach towards operationalizing resilience would very much strengthen the manuscript and substantiate the findings. I would therefore suggest to derive several residual metrics indicative of cognitive reserve using other cognitive domains and pathologic markers, and then replicate the analyses presented in the paper. For example, years of formal education or executive function from the trail-making test can be used as another indicator of cognitive performance. In the MRI data, hippocampal atrophy, white matter lesions on the structural MRI data, and ventricular volumes can be used as alternative pathologic markers. Previous research demonstrated that these MRI markers showed stronger associations with cognitive decline, progression towards dementia, and dementia risk. They are therefore better outcome variables for the regression model as the MRI total brain volume, and may be more sensitive a neuronal correlate of cognitive decline. Adding these additional regression models would help to cross-validate the cognitive resilience construct and strengthen the findings of the present study.

We thank the reviewer for this point.

Although validation of the residual method was not a specific aim of this work, we do appreciate that this method is somewhat new and uncommon, and many readers may be unfamiliar with the metrics and concepts. Therefore, we added two additional metrics for cognitive reserve to the analyses, and described differences in the results across the three metrics. Please note that we were not able “validate” the metrics, because there is no gold standard measure (or *any* direct measure) for cognitive reserve, so there is no way to directly evaluate these metrics for validity.

The metrics we added were: (A) COWA fas test score regressed against white matter hyperintensity volume (standardized to intracranial, IC volume), representing a vascular-dominant pathological process and vascular dementia-sensitive symptomology (fluency, executive function); and (B) CVLT-II short form long delay free recall score regressed against hippocampal volume (per IC volume), representing neurodegenerative dominant process and neurodegenerative sensitive symptomology (memory). These joined the WAIS digit symbol coding test regressed against overall brain volume (per IC volume), representing global loss in pathology and global loss in function (processing speed).

We did evaluate years of education as a determinant for development of cognitive reserve, but not as a metric representing cognitive reserve.

There may be research demonstrating associations of HC volume loss, WMH volume, and ventricle enlargement as strong predictors of cognitive loss in other populations, but please note that this population is distinct with substantive differences. Prior research has shown that processing speed (WAIS DSST) was the most sensitive test to any MRI defined features, and that overall brain volume, HC volume, and WMH volume (in descending order) had the strongest associations with cognitive test features. We have added some of this context to our Background (page 2-3), as well as the new analyses described throughout the paper in Methods, Results, Discussion, and Tables/Figures.

Minor

The term "American Indians" should be avoided to follow APA 7 guidelines on the usage of bias-free language in research. Bias-free alternative ethnic descriptors are "Native Americans", "Native North Americans" or "indigenous people of (North) America". The authors justify the use of the term "American Indian" [...] in accordance with contemporary US legal structures." However, this justification is insufficient for a medical science publication because US legal terms are not reflective of bias-free language use and the current knowledge about ethnicity in a research context. For example, intersectionality is an important concept in research on ethnic groups, but it does not exist in the US legal system as seen in the *DeGraffenreid v. General Motors* case. The language used by contemporary US legal structures is not suitable for research publications. More information on bias-free language can be found here: [] refer to an avoid the term "Indian".

We thank the reviewer for their consideration of community perspectives in research terminology.

Page 144 of the 7th edition of APA does not say that the term "American Indian" is inappropriate; it says that other terms may be more preferable, although it does not justify this claim. Please note that the APA does not set guidance and terminology preferences for American Indian and other Indigenous peoples, and so recommendations on the basis of APA guidance may be inappropriate for describing community-based research in these settings.

We used the term "American Indian" not only because it aligns with US legal structures, but also based on specific direction from our study partner Tribal communities and our Community Advisory Board. Terminology preferences do vary by Tribe, community, individual, and country-- but in general this term ("American Indian") is considered acceptable in the United States; is generally the preferred term when individual Tribe or Tribal nations cannot be directly named; and is provided as guidance by nationally-representative, self-governing, Indigenous-led organizations such as the National Congress of American Indians. Terms preferred in other countries or by other Peoples do not change the preferences of US-based American Indian Tribes, who have the final say in representation of their identities.

We have clarified justification of use of this term in our text, so that use and guidance for this terminology from our Tribal partners and our CAB is made more explicit. (Page 3)

The Wechsler Adult Intelligence Scale is not a culture-free intelligence test and may be biased when carrying out research in Native Americans. Since all participants were Native Americans, this bias is unlikely to affect the findings, because the bias mainly becomes apparent when comparing Native Americans with European Americans. Nevertheless, the culture bias of the test should be mentioned in the limitations section along with the need for further research using culture-free tests.

We thank the reviewer for this comment. The WAIS digit symbol coding test is the test that was used in our battery; not the entire WAIS IQ battery. We have clarified this point in the Methods and Tables (page 3). We also added a statement to Limitations related to generalizability due to score interpretability and cultural validity. (Page 8).

Estimates of reliability (internal consistency): A report of the internal consistency of the WAIS scale and the other psychometric data used is missing. At least, Cronbach's alpha should be reported at the beginning of the results section. It is important to know the reliability estimates from the sample because the tests were used in the Native American population but constructed and validated in the general population. Similarly, an internal consistency metric for the CESD should be reported due to differences in depression across various ethnic groups.

We thank the reviewer for this question. For the cognitive tests, we have done independent psychometric analysis for this population for 3MSE, MoCA, COWA; psychometric evaluations for WAIS, CVLT, and remainder of UDS form C2 battery are still underway. Internal consistency-reliability by calculating omega

coefficient. Omega values >0.90 are considered adequate for individual decision-making and >0.8 adequate for research (Nunnally, 1994). For 3MSE, MoCA, and COWA omega coefficients were 0.9 for the overall score suggesting excellent interpretability of the summary score in both research and clinic settings; however, sub-scores or individual domain scores were <0.8, indicating poor statistical support for interpretation of individual domain scores. (Suchy-Dicey et al JINS 2024, Suchy-Dicey et al, Assessment 2024, Suchy-Dicey et al Assessment, 2023). We have added a statement to this effect, with references to our prior papers, in the Methods. (page 3-4).

BMI: The BMI was categorized into normal, overweight, and obese. It would be interesting to know if any participants were underweight with a BMI less than 18.5. Previous research showed that being underweight was associated with cognitive impairment in late life.

We thank the reviewer for this question. There were 5 participants with BMI<20, and 2 with BMI<18.5 (17.5 and 18.1). This is too few for an independent category, but we agree that excluding them is not ideal, so we re-coded the variable as an indicator: overweight/obese vs not (normal/underweight). In prior research, OW/obese were (somewhat paradoxically) detected to have lower cognitive decline than normal and underweight groups, so the inclusion/exclusion of this group would not be expected to materially change the analyses.

Alcohol: The plot (Supplementary Figure 4) shows alcohol has a skewed distribution with only a few participants consuming more than 7 drinks per day. This limits the generalizability of the non-significant effect of alcohol consumption on cognitive reserve and should be mentioned as a limitation.

We thank the reviewer for this point. Very few elders in this population drink, and those who do tend to drink little. Almost none fulfill the criteria of “heavy drinker” as defined by the Lancet Commission, of 21 drinks per week. We added context to tables that mean alcohol use (continuous drinks per day) was calculated among users, and provided the count of users, to highlight this skew. Alcohol was not associated with the 3 metrics, either in degree or in power, which is consistent with prior findings in this population and likely is due to the very low proportion of drinking habits. We added a statement in Discussion offering a potential interpretation of some null associations (eg for alcohol) reflecting low variance in the measure for the population observed (page 7).

“Possibly other measurements of cognitive reserve, either using different cognitive and pathologic measures or different mathematic constructs, may provide a better map for this esoteric construct.” “Esoteric construct” is not a good word choice here, I would replace this with “operationalizing this unobserved, latent construct.”

We thank the reviewer. We have made many changes to the text, and this statement is no longer included.

“On the other hand, better cognitive reserve may be either a contributor or an effect of disease comorbidities such as diabetes and depression.” Should this sentence read: “lower cognitive reserve.”

We thank the reviewer; we have adjusted the sentence to read “differential cognitive reserve”, so that the directionality is not pre-specified. (page 8)

Table 4: The individual component paths of the mediation model should be reported as well: The path from the exposure to the mediator (Diabetes V1 à Cognitive V1) and mediator to the outcome (Cognitive reserve V1 -> Composite cognitive score V2). These paths can explain why the mediation was not significant.

We thank the reviewer. We have added the total effect, direct effect, and indirect effect for the mediation models (as well as the proportion mediated). We appreciate this suggestion, and agree that Table 4 is more complete with these additions.

Reviewer #2 (Remarks to the Author):

ABSTRACT

1. In lines 3-4, the authors write, "However cognitive resilience has not been measured or evaluated in American Indian elders." Please clearly state the purpose of the investigation, e.g., "The purpose of this investigation is to evaluate factors associated with cognitive reserve or cognitive resilience in American Indian elders."

We thank the reviewer. We did include the statement "This study aimed to develop understanding of these metrics in this uniquely burdened population", referring to the terms previously defined. However, we also added substantial material to clarify and justify the metrics, and why they were included. (page 2-3).

2. Cognitive reserve and cognitive resilience are conceptually different but the authors seem to use them interchangeably. Please use one term to avoid confusing readers.

We thank the reviewer for this point. We do use the term "resilience" to introduce the concept. However, we have clarified the terminology, to avoid use of the term "resilience" except for broad conceptual discussion. We have used the term "cognitive reserve" when discussing concept of continued cognitive health despite evidence of accumulating pathology.

INTRODUCTION/BACKGROUND

1. The state of the field is clearly expressed in this section, which justifies the importance of the research question.

We thank the reviewer.

2. The concepts of cognitive reserve and resilience overlap, so this section warrants more explanation to avoid confusing readers.

We thank the reviewer. We did move some of the background to the introduction, as requested in a later comment, to better introduce the concepts and field and to orient the reader to the concepts and terms more clearly. (Page 2).

3. Please consider clearly defining cognitive reserve and cognitive resilience per the Collaboratory on Research Definitions for Reserve and Resilience on Aging (PMID: 36653245) earlier in the manuscript. It is referenced in the discussion. An upfront and early delineation of the concepts can improve the reader's comprehension of the paper.

We thank the reviewer. We do define these terms in a manner consistent with the Collaboratory group and refer also to the citation provided, and also moved some of the discussion of terms up to the Introduction, as requested. (Page 2)

4. The introduction mentions that American Indian persons are at increased AD risk, but the authors do not quantify that risk with a rate.

We thank the reviewer. We added additional citations to reflect AD and related dementias. We do not include rate because incident rate is not known for this population, and the values for “vascular disease, Alzheimer’s disease, and related dementias” (as discussed) vary, so this would be a lengthy recitation of previously published numbers. However, risk for dementia is higher than for other groups, with >50% of American Indians age >70 adjudicated as impaired by expert panel case review and consensus.

5. If persons aged 80 and older are in the sample, it would be interesting to use cognitive testing to determine what proportion meets criteria consistent with super-agers.

We thank the reviewer for this suggestion. There are n=121 (91 F, 30 M) aged >80 years in the Visit 1 examination cohort. Standard score cutoffs to discriminate cognitive healthy from impaired are not appropriate for this population, due to down/left-shifted score distributions and poor discriminant validity of test scores for cognitive status. Thus, it would not be possible to readily ascertain a number who fit the criteria of “super ager”. However, evaluation of super-aging is an excellent future direction, and is part of ongoing research in this cohort study. We have added this to the future directions in Discussion (page 8).

METHODS

1. The selection of study participants and study variables utilized for the investigation are clear.

We thank the reviewer.

2. Is using regression of WAIS-DCS scores over standardized total brain volume estimates to operationalize cognitive reserve a proven approach? The paper cited used and validated the method in the context of cognitive resilience. Again, be cautious about how the terms “reserve” and “resilience” are used. Please make this point clear so as not to confuse readers. If the approach represents a novel use for cognitive reserve, is data supporting its validity in that context?

We thank the reviewer for this question.

We do appreciate that this method is somewhat new and “unproven”, and many readers may be unfamiliar with the metrics and concepts. Therefore, we added two additional metrics for cognitive reserve to the analyses, and described differences in the results across the three metrics. We were not able “validate” the metrics, because there is no gold standard measure, or *any* direct measure, for cognitive reserve; rather, all are hypothetical representations of the conceptual construct. We included more material in the Introduction to make these points more clear.

The metrics we added were: (A) COWA fas test score regressed against white matter hyperintensity volume (standardized to intracranial, IC volume), representing a vascular-dominant pathological process and vascular dementia-sensitive symptomology (fluency, executive function); and (B) CVLT-II short form long delay free recall score regressed against hippocampal volume (per IC volume), representing neurodegenerative dominant process and neurodegenerative sensitive symptomology (memory). These joined the WAIS digit symbol coding test regressed against overall brain volume (per IC volume), representing global loss in pathology and global loss in function (processing speed).

We did evaluate years of education as a determinant for development of cognitive reserve, but not as a metric representing cognitive reserve.

Prior research has shown that processing speed (WAIS DSST) was the most sensitive test to any MRI defined features, and that overall brain volume, HC volume, and WMH volume (in descending order) had the strongest associations with cognitive test features. We have added some of this context to our Background (page 2-3), as well as the new analyses described throughout the paper in Methods, Results, Discussion, and Tables/Figures.

RESULTS

1. The data are presented appropriately, but it needs to be clarified which results are statistically significant in Table 1.

We thank the reviewer for this question. We do not generally include P-values in Table 1, unless there is an inferential purpose to the table. In this paper, we agree that we do make the case that cognitive reserve metrics are similar/different between Visits; therefore, we have added P-values to the table, as requested.

2. Define what “more formal education” means (line 121). Are you referring to years of education?

We thank the reviewer. Yes; and we have added “years of” in order to avoid confusion with a scale of more or less formalized education. (page 5)

3. Please use spelling and grammar-checking software to detect misspelled words, “Both diabetes and depression had significant direct effect ($P < 0.0001$)...” (line 145).

We thank the reviewer. Our spell check did indicate accuracy of spelling; thus, for some reason, this tool in Word/Office is not detecting these mistakes. We have tried to identify typographical errors manually, but if there are any that we overlooked, we are happy to correct them.

DISCUSSION

1. Be clear that cognitive reserve and cognitive resilience are conceptually different. In lines 158 to 160, the following sentence suggests they are interchangeable: “Cognitive reserve, or cognitive resilience to biological or pathological changes, is challenging to study—first because as a theoretical, and direct measures are not available, so operationalization of proxy and latent variables is necessary.”

We thank the reviewer. We appreciate this is a challenging area, and that consistency is important for field-wide clarity. As noted above, we have adjusted the use of these terms to try to avoid their confusion and conflation.

2. If the sample reduction between Visit 1 and Visit 2 is unrelated to selective survival, please comment on other possibilities.

We thank the reviewer for this question. It may be helpful to distinguish between survival and selective survival. The losses between our examination visits are indeed primarily due to mortality / survival; however, there is not strong evidence that the variables of interest (cognitive reserve) differ over time due to any undue bias / selective survival. In fact these measures don't appear to differ over time at all—the three metrics were similar between Visit 1 and Visit 2. We amended the statement on selection bias to more clearly reflect this. (Page 8)

OVERALL

1. The authors address an important topic in cognitive, i.e., factors associated with resilience in an underrepresented research population.

We thank the reviewer

2. Exploring cognitive risk and protective factors in diverse samples can yield critical insights into disease manifestation, treatment, and prevention. Please consider citing the Alzheimer's Association Facts and Figures reports that discuss ADRD research samples that do not match the country's general population to highlight your work's importance further.

We thank the reviewer for this suggestion. We have added citation to the AA Facts & Figures 2024 document in our discussion section related to social determinants which influence development of cognitive reserve, such as early education and social engagement, which are the main topics of the present manuscript. Other papers from our research program, such as on prevalence of dementia in this cohort (Suchy-Dicey et al Alzheimer's & Dementia 2024), do also cite the same AA F&F 2024 document, vis a vis the epidemiology of dementia and population cross-comparisons.

3. The longitudinal design is a strength of this work.

We thank the reviewer .

4. The authors alternate between adjectival and noun forms, i.e., American Indians vs. American Indian elders. Use the same form per the journal's guidance for consistency.

We thank the reviewer. We have clarified the use of collective (American Indians) and group (elders/older adults) nouns throughout. The exact same term cannot be used in all instances because the meaning (American Indians vs American Indian older adults) is not the same, but we have ensured that the use of terms is consistent and intentional.

5. Overall, the paper is important and timely to the field. There are a few grammatical errors. Spelling and grammar-checking software is a consideration for catching errors and improving overall readability. The word, cognitive, is misspelled in the abstract: "Analyses examined residuals of cogntiive reserve against sociodemographic, clinical, and longitudinal cognitive data in causal mediation models."

We thank the reviewer. We have corrected all instance of misspelled words that we were able to detect; if we overlooked any, we will be happy to review.